# Systematic examination of a heart failure risk prediction tool: The pooled cohort equations to prevent heart failure

Aakash Bavishi[1], Donald M. Lloyd-Jones[2,3], Hongyan Ning[2], Thanh Huyen T. Vu[2], Clyde W. Yancy[3], Sanjiv J. Shah[3], Mercedes Carnethon[2], Sadiya S. Khan [2,3]*

1 Department of Medicine, Northwestern University Feinberg School of Medicine, Chicago, Illinois, United States of America, 2 Department of Preventive Medicine, Northwestern University Feinberg School of Medicine, Chicago, Illinois, United States of America, 3 Division of Cardiology, Department of Medicine, Northwestern University Feinberg School of Medicine, Chicago, Illinois, United States of America

* s-khan-1@northwestern.edu

**Data Availability Statement:** All relevant data are within the paper and its Supporting Information files.

## Abstract

Identification of individuals at risk for heart failure is needed to deliver targeted preventive strategies and maximize net benefit of interventions. To examine the clinical utility of the recently published heart failure-specific risk prediction model, the Pooled Cohort Equations to Prevent Heart Failure, we sought to demonstrate the range of risk values associated with diverse risk factor combinations in White and Black men and women. We varied individual risk factors while holding the other risk factors constant at age-adjusted national mean values for risk factors in each race-sex and age group. We also examined multiple combinations of risk factor levels and examined the range of predicted 10-year heart failure risk using the Pooled Cohort Equations to Prevent Heart Failure risk tool. Ten-year predicted heart failure risk varied widely for each race-sex group across a range of ages and risk factor scenarios. For example, predicted 10-year heart failure risk in a hypothetical 40 year old varied from 0.1% to 9.7% in a White man, 0.5% to 12.3% in a Black man, <0.1% to 9.3% in a White woman, and 0.2% to 28.0% in a Black woman. Higher risk factor burden (e.g. diabetes and hypertension requiring treatment) consistently drove higher risk estimates in all race-sex groups and across all ages. Our analysis highlights the importance of a race and sex-specific multivariable risk prediction model for heart failure to personalize the clinician-patient discussion, inform future practice guidelines, and provide a framework for future risk-based prevention trials for heart failure.

## Introduction

Heart failure (HF) currently affects 6.2 million adults in the United States (US), with projections that over 8 million people will have HF by 2030 [1,2]. HF places a significant burden on the healthcare system, with direct annual costs of $11 billion in 2014, and projected direct annual costs of $53 billion dollars by 2030 in the US [1,3]. Even more troubling are recent data that identify worsening trends in age-adjusted mortality rates due to HF [4,5]. As a result, HF

**Funding:** Research reported in this publication was supported, in part, by the National Institutes of Health's National Center for Advancing Translational Sciences, Grant Number KL2TR001424 and the American Heart Association (#19TPA34890060) to SSK. The content is solely the responsibility of the authors and does not necessarily represent the official views of the National Institutes of Health. Sanjiv J. Shah is supported by grants from the National Institutes of Health (NIH; R01 HL107577, R01 HL127028, and R01 HL140731) and the American Heart Association (AHA; #16SFRN28780016 and #15CVGPSD27260148). The funders had no role in study design, data collection and analysis, decision to publish, or preparation of the manuscript.

**Competing interests:** The authors have declared that no competing interests exist.

guidelines by American College of Cardiology/American Heart Association (ACC/AHA) and European Society of Cardiology (ESC) highlight the necessity to identify individuals at increased risk of developing HF and emphasize prevention of HF with implementation of targeted intervention strategies [6,7].

Risk-based prevention, in which the intensity of the intervention matches the estimated absolute risk, is widely accepted in primary prevention of atherosclerotic cardiovascular disease (ASCVD), but no such paradigm currently exists for HF [8–12]. Cardiovascular risk estimation tools that integrate multiple risk factors to quantify absolute risk have been demonstrated to be superior in classification of individuals compared with risk factor counting or subjective clinician assessment alone for ASCVD [13,14]. For HF-specific risk assessment, we recently published the Pooled Cohort equations to Prevent Heart Failure (PCP-HF) using pooled individual-level data from seven diverse population-based cohorts to derive and validate sex- and race-specific estimates of 10-year risk of incident HF [15].

Our prior work investigating the intrinsic properties of the Framingham Heart Study General Cardiovascular Risk Profile and the ACC/AHA Pooled Cohort Equations for ASCVD highlighted the importance of this approach in identifying inherent limitations to population-based prediction models when applied to individuals [16,17]. With the recent publication of the PCP-HF tool (available online at http://pcphf.org), we aimed to examine the intrinsic properties of this HF-specific risk model to demonstrate its properties and how it may be useful to clinicians to guide risk communication discussions. Therefore, the objective of this study is to systematically evaluate the PCP-HF risk prediction tool to determine the impact of varying risk factor levels and combinations in different age, sex, and race groups [16].

## Materials and methods

### Risk prediction model: The PCP-HF tool

Briefly, the PCP-HF tool incorporates age, SBP (systolic blood pressure), fasting glucose, BMI (body mass index), TC (total cholesterol), HDL-C (high density lipoprotein-cholesterol), smoking status, use of antihypertensive medication, use of diabetes mellitus (DM) medications, and QRS duration on electrocardiogram into multi-variable Cox proportional hazard regression equations to predict 10-year absolute risk for incident HF as previously published (S1 Fig) [15].

We created spreadsheets for each race-sex group, incorporating the coefficients for each risk factor and interaction term based on the published PCP-HF tool to create risk estimates for a hypothetical White man, Black man, White woman, and Black woman from 30 to 70 years in 10-year age increments (S1 Table). This approach does not examine risk for a specific individual or specific populations; rather, it varies risk factor levels individually and in pre-specified combinations in a systematic manner to examine the range of outputs for the 10-year HF risk prediction tool.

### Variation of single risk factor

We varied each individual risk factor while holding the other risk factor levels constant at age-adjusted national mean values in order to compare the effects of single risk factors on 10-year predicted HF risk in all sex and race subgroups. We used data from noninstitutionalized, non-pregnant adults aged 30–80 years in each sex and race group who were free from CVD and participated in the National Health and Nutrition Examination Survey (NHANES) 2011–2016 cycles to calculate age-adjusted national mean values for risk factor levels (Table 1). We used the surveyfreq, surveymeans, surveyreg, and surveylogistic procedures in SAS 9.4 (SAS Institute, Inc, Cary, NC), which permit calculation of standard error for samples that include a

**Table 1. Age-adjusted mean national levels of risk factors by race-sex group.**

|  | White Men | Black Men | White Women | Black Women |
|---|---|---|---|---|
| Untreated SBP (mmHg) | 122(0.4) | 129(0.6) | 118(0.4) | 123(0.8) |
| Treated SBP (mmHg) | 129(0.8) | 136(1.0) | 131(0.8) | 137(1.0) |
| Untreated Glucose(mg/dL) | 103(0.5) | 103 (1.0) | 99(0.4) | 102(1.2) |
| Treated Glucose(mg/dL) | 160(4.6) | 157(10.6) | 160(6.9) | 150(8.3) |
| TC (mg/dL) | 192(1) | 187(1.5) | 201(1.0) | 191(1.4) |
| HDL-C(mg/dL) | 48(0.4) | 51(0.4) | 59(0.4) | 59(0.4) |
| BMI(kg/m$^2$) | 29.3(0.1) | 29.8(0.2) | 29.4(0.2) | 33.2(0.24) |

BMI, body mass index; HDL-C, high-density lipoprotein cholesterol; SBP, systolic blood pressure; TC, total cholesterol.

[a] Values are mean (standard error) and were calculated from the 2011–2016 National Health and Nutrition Examination Surveys (NHANES).

complex weighting scheme to generate point estimates. We used different mean SBP values and fasting glucose values for those who were treated and not treated with anti-hypertensive therapy or DM therapy. Mean QRS duration for all race-sex groups was set to 90ms based on the mean value observed in the derivation cohort. For these analyses, we used non-smoking as the default input because it is the normative state in the population. Furthermore, we assumed untreated SBP and untreated fasting glucose as the default when other risk factors were being varied.

We varied levels of individual risk factors as follows: SBP from 90 to 190 mmHg in intervals of 20 mmHg; fasting glucose from 80 to 200mg/dL in intervals of 20mg/dl; BMI from 20 to 45 kg/m$^2$ in intervals of 5 kg/m$^2$; TC from 120 to 300 mg/dl in increments of 30mg/dl; HDL-C from 30 to 60 mg/dl in intervals of 5mg/dl; and QRS duration from 80 to 180ms in intervals of 20ms. These ranges were selected based on clinically relevant values that were included in the derivation cohort, given the known limitations with extreme outliers of risk factor levels.

We also calculated distinct 10-year predicted HF risk estimates with and without antihypertensive therapy and with and without DM therapy, holding all other risk factors constant at age-adjusted national mean values derived for treated US adults for each race-sex group. For example, to determine the effect of BMI on 10-year predicted HF risk in a 60-year old Black man, we set untreated SBP to 129 mm Hg, untreated fasting glucose to 103 mg/dL, TC to 187 mg/dL, HDL-C to 51 mg/dL, and QRS duration to 90 ms, selected non-smoking status, and varied BMI from 20 to 45 kg/m$^2$. For all analyses, we calculated 10-year predicted HF risk for each race-sex group.

### Variation of multiple risk factors

We examined the effects of different risk factor combinations on 10-year predicted HF risk. We chose value ranges with intervals approximating 1 standard deviation in the population of US adults (SBP) or clinically meaningful values (fasting glucose, BMI). For SBP, values included were 120 mm Hg, 140 mm Hg, and 160 mm Hg. Fasting glucose values included were 100 mg/dL, 150 mg/dL, and 200 mg/dL. For BMI, values included were 25 kg/m$^2$, 30 kg/m$^2$, and 35 kg/m$^2$. We varied the categorical variables of smoking status, antihypertensive therapy and DM therapy for all risk factor combinations. We examined ages for several intervals but only display results of each sex and race subgroup at 60 years old. We chose this age given that prevalence of HF increases substantially after 60 years of age and we judged that it would offer the most relevant risk factor combinations in a theoretical at-risk patient, understanding that the absolute predicted HF risk varies significantly with age.

## Results

### Effects of varying age with other risk factors held constant

When all other risk factors were held constant at the age-adjusted national mean level for each race-sex group, 10-year predicted HF risk was substantially higher with older age (Fig 1). In individuals requiring treatment for both HTN and DM was associated with a higher predicted 10-year HF risk for theoretical individuals of all race-sex groups, with the effect being most significant at older ages. For example, at age 60 a hypothetical White man not requiring treatment for HTN or DM with an average risk factor profile (age-adjusted national mean values) would have a 10-year predicted HF risk of 2.4%. In comparison, a hypothetical 60 year old White man with identical risk factors requiring treatment for both HTN and DM would have a 4.6-fold higher predicted 10-year HF risk (11.1%). Similarly, 10-year predicted HF risk would be 3.9-fold higher in a Black man (9.4% vs 2.4%), 4.6-fold higher in a White woman (7.9% vs 1.7%), and 5.6 fold higher (9.5% vs 1.7%) in a Black woman requiring treatment for both HTN and DM to achieve the same risk factor levels compared to a theoretical individual with the same risk factor values not on treatment for HTN or DM.

Ten-year predicted risk for heart failure (HF) for a hypothetical White and Black man and woman at interval selected ages, with risk factors held constant at approximate age-adjusted national means (among nonsmokers). (A) Ten-year HF risk estimates for those not taking antihypertensive or diabetes medications. (B) Ten-year HF risk estimates for those taking antihypertensive but not diabetes medications. (C) Ten-year HF risk estimates for those taking diabetes but not antihypertensive medications. (D) Ten-year HF risk estimates for those taking antihypertensive and diabetes medications. BP, blood pressure; DM, diabetes mellitus.

### Effect of varying single risk factors with other risk factors held constant

Estimated 10-year HF risks for a Black and White man and woman after individually varying untreated SBP, treated SBP, untreated glucose, treated glucose, BMI, TC, HDL-C, and QRS duration are shown in Figs 2–5 and S2 and S3 Figs; for each analysis, all other risk factors are held constant at age-adjusted mean levels. At all ages in different race-sex groups, current smoking was associated with an approximately 2-fold higher 10-year predicted HF risk. Requiring treatment for HTN was also associated with a 1.5 to 2-fold higher 10-year predicted HF risk at all ages in all race-sex groups. Requiring treatment for DM was associated with a 1.3 to 2.3-fold higher 10-year predicted HF risk.

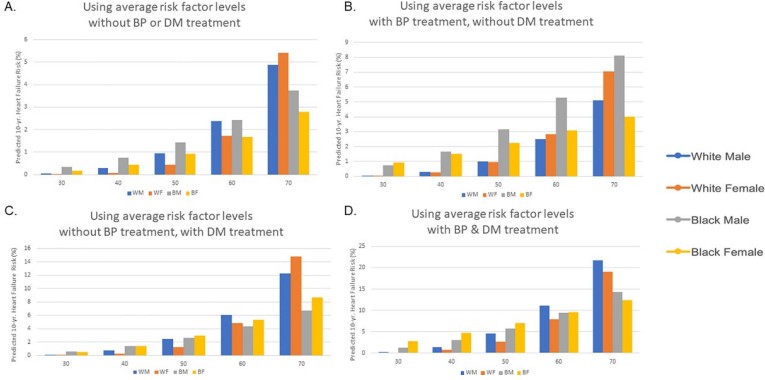

**Fig 1. Ten-year predicted heart failure risk by race-sex group.**

## White Man

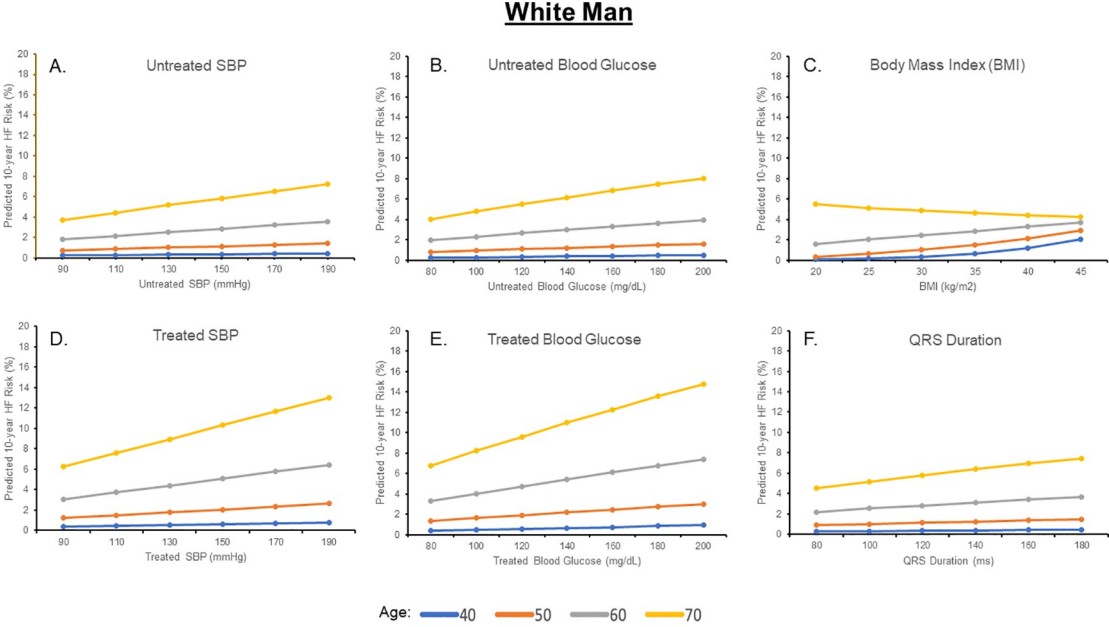

**Fig 2. Ten-year predicted heart failure risk by varying single risk factor levels in a hypothetical white man.**

Ten-year predicted risks for heart failure by varying levels of single risk factors in a hypothetical White man at selected ages, with other risk factors held constant at approximate age-adjusted national means (including nonsmoking). SBP, systolic blood pressure.

Ten-year predicted risks for heart failure by varying levels of single risk factors in a hypothetical Black man at selected ages, with other risk factors held constant at approximate age-adjusted national means (including nonsmoking). SBP, systolic blood pressure.

## Black Man

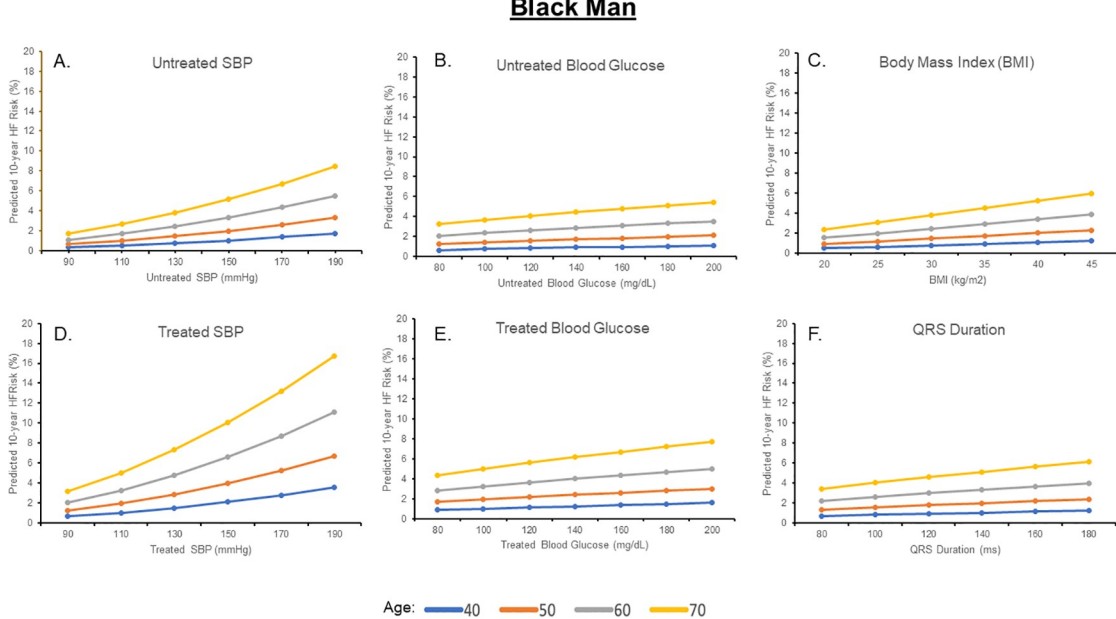

**Fig 3. Ten-year predicted heart failure risk by varying single risk factor levels in a hypothetical black man.**

## White Woman

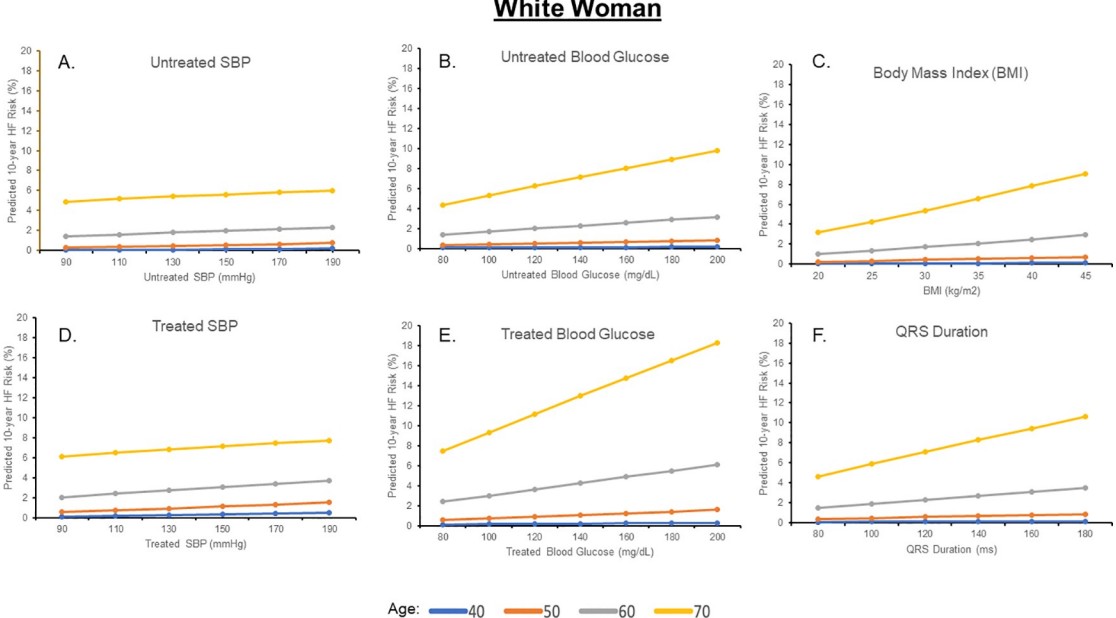

**Fig 4. Ten-year predicted heart failure risk by varying single risk factor levels in a hypothetical white woman.**

Ten-year predicted risks for heart failure by varying levels of single risk factors in a hypothetical White woman at selected ages, with other risk factors held constant at approximate age-adjusted national means (including nonsmoking). SBP, systolic blood pressure.

Ten-year predicted risks for heart failure by varying levels of single risk factors in a hypothetical Black woman at selected ages, with other risk factors held constant at approximate age-adjusted national means (including nonsmoking). SBP, systolic blood pressure.

## Black Woman

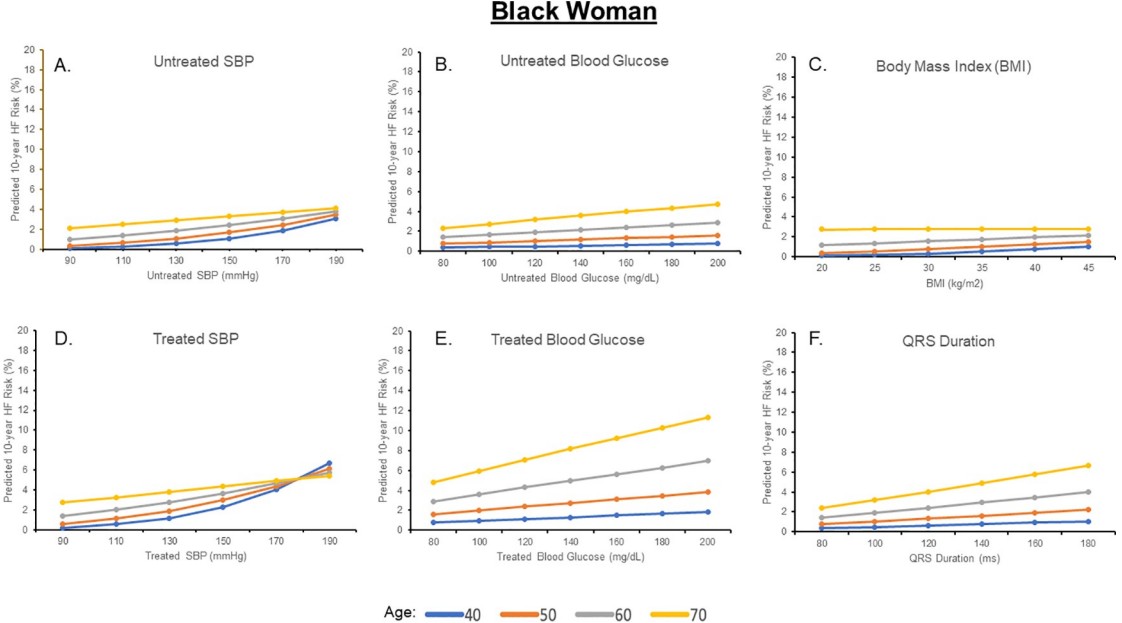

**Fig 5. Ten-year predicted heart failure risk by varying single risk factor levels in a hypothetical black woman.**

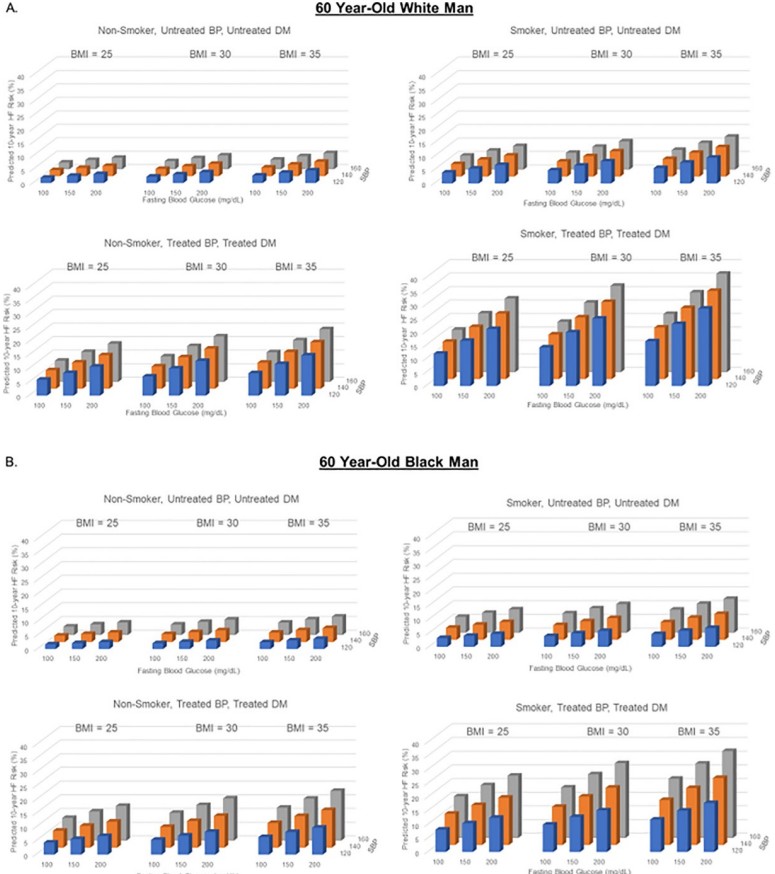

**Fig 6. Ten-year predicted heart failure risk by varying multiple risk factors at 60 years of age in a hypothetical white and black man.**

## Effect of varying multiple risk factors simultaneously

Ten-year predicted HF risk with simultaneous variation of multiple risk factors is demonstrated for a hypothetical 60-year-old of each sex and race group in Figs 6 and 7 and S4–S6 Figs. Ten-year predicted HF risk in a hypothetical 40-year-old varied widely from 0.1% to 9.7% in a White man, 0.5% to 12.3% in a Black man, 0.0% to 9.3% in a White woman, and 0.2% to 28.0% in a Black woman.

Ten-year predicted risks for heart failure by varying levels of multiple risk factors in a hypothetical White man (A) and Black man (B) at 60 years of age. BMI = body mass index; BP, blood pressure; DM, diabetes mellitus; SBP, systolic blood pressure.

Ten-year predicted risks for heart failure by varying levels of multiple risk factors in a hypothetical White woman (A) and Black woman (B) at 60 years of age. BMI, body mass index; BP, blood pressure; DM, diabetes mellitus; SBP, systolic blood pressure.

## Effect of changing QRS duration

Ten-year predicted HF risk with QRS varying from 90ms to 120ms for a hypothetical 60-year-old in each race-sex group with moderate risk factors is demonstrated in S7 Fig. Comparison of ten-year predicted HF risk with QRS 100ms or 110ms is shown at interval ages for each race-sex group in S8 Fig.

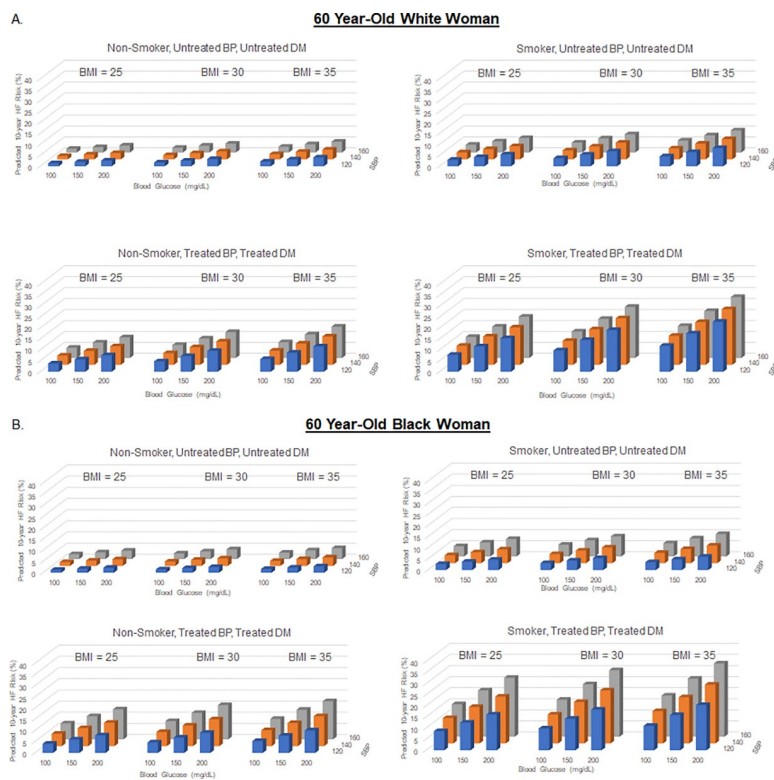

**Fig 7. Ten-year predicted heart failure risk by varying multiple risk factors at 60 years of age in a hypothetical white and black woman.**

## Discussion

### Principal findings

In this study, we systematically evaluated the intrinsic properties of the recently published PCP-HF risk prediction tool with various risk factor combinations. This demonstrates that a wide range of risk values are associated with diverse risk factor combinations, which cannot be easily interpreted from the coefficients alone, given the complex associations and interactions of risk factors that are integrated in the model. While age is a known major driver of predicted 10-year HF risk in this model, several variables in isolation or combination can significantly influence 10-year predicted HF risk even at the same age. Separate race- and sex-specific coefficients revealed important differences in predicted risk for HF across race and sex groups with similar risk factor profiles.

The systematic examination of the multivariable PCP-HF tool confirms previously demonstrated risk factors of HF [18]. The previously validated components of the PCP-HF tool were selected based on their individual associations with incident HF and reflect causal factors that contribute to HF risk (obesity, diabetes, hypertension, hyperlipidemia, and smoking) as well as subclinical surrogates of HF risk (QRS duration) [19,20]. For example, higher levels of SBP were associated with greater predicted risk of HF in all race-sex groups. While higher BMI values were generally associated with greater predicted risk of HF, there was an inverse association between BMI and predicted risk of HF in the theoretic risk curve for a 70-year old White man. One potential explanation may be the controversial obesity paradox in which patients with obesity with risk factors for HF such as HTN or DM paradoxically had better outcomes in terms of mortality when compared with patients in lower BMI strata [21–24]. Specifically,

among older adults with obesity, there are several explanations for this phenomenon, including a greater competing risk of mortality from non-HF causes, muscle wasting and sarcopenia, as well as malnutrition or development of cachexia [25]. Finally, BMI is a poorer measure of adiposity in white men and may reflect greater burden of frailty compared with other race-sex groups [26].

Race-sex differences in the relationship between established risk factors and predicted risk were apparent. For example, elevated levels of SBP despite anti-hypertensive treatment in a 70-year-old has much more influence on 10-year predicted HF risk for a Black man than it does on a White man. When multiple risk factors were varied, race-sex interactions continued to have significant effects on 10-year predicted HF risk. These findings are also consistent with observed racial disparities in incidence of HF reported in several contemporary community-based cohort studies [27–30]. Compared with other HF risk prediction tools, the use of sex- and race-specific coefficients specifically enhances identification of at-risk women and Black adults.

## Risk-based prevention to reduce burden of HF

Over the past 4 decades, risk-based strategies for ASCVD prevention have centered upon the principle of matching the intensity of prevention strategies with the estimated absolute risk. Several risk prediction tools have been developed to assist in quantifying risk estimates for atherosclerotic CVD and are in widespread clinical use to guide risk-based prevention strategies [10–12,31–33]. Comprehensive CVD risk tools including ASCVD and HF outcomes have been demonstrated to have poor calibration when assessing HF-specific risk, which may be related to distinct pathophysiologic mechanisms contributing to the development of HF compared with ASCVD [15,34–36]. Available risk estimation tools for HF have typically lacked extensive external validation, broad applicability to a general population, and use of clinical parameters readily available to a primary care physician. Therefore, translation of a comprehensive risk-based approach in the primary prevention of HF has been lacking to-date. Implementation of HF-specific risk tools may facilitate preventive efforts, reduce incidence of HF, and improve cardiovascular outcomes in the population.

## Emerging risk-based prevention strategies for HF

One emerging strategy for targeted prevention of HF may be the use of sodium-glucose cotransporter 2 inhibitors (SGLT2i) in patients with and without diabetes. In data from the Empagliflozin Cardiovascular Outcome Event Trial in Type 2 Diabetes Mellitus Patients–Removing Excess Glucose trial and the Empagliflozin Comparative Effectiveness and Safety Study, participants without pre-existing HF at baseline experienced significantly fewer HF hospitalizations with empagliflozin by approximately 50% compared with placebo [37,38]. In the Dapagliflozin Effect on Cardiovascular Events–Thrombolysis in Myocardial Infarction trial, dapagliflozin was also shown to reduce risk of cardiovascular death or hospitalization for HF, (HR 0.84 [0.72–0.99]) [39]. The Dapagliflozin and Heart Failure with Reduced Ejection Fraction trial demonstrated the efficacy of dapagliflozin in non-diabetics in reducing a composite outcome of HF hospitalizations, requirement of intravenous diuretic therapy, and death from cardiovascular cause (HR 0.73 [0.60–0.88]) [40]. Preclinical models with SGLT2i have demonstrated favorable cardiac remodeling (e.g. improvement in left ventricular hypertrophy, systolic and diastolic function) in both non-diabetic rat and porcine models [41,42]. Risk-based randomized control trials to further investigate the efficacy of SGLT2i in the primary prevention of HF among patients with and without diabetes are needed and may inform novel patient populations who may benefit [43,44]. Use of the PCP-HF tool can also be applied to enrich

clinical trial enrollment by focusing on at-risk individuals to investigate novel therapies in a targeted manner to broaden risk-based prevention interventions.

The importance of intensive blood pressure lowering to reduce risk of HF was demonstrated in the Systolic Blood Pressure Reduction Intervention Trial in which participants 50 years of age or older with HTN had a 36% lower rate of HF in the intervention arm (SBP target <120 mmHg) compared with those who were in the control arm (SBP target <140mmHg) [45]. Similar to the implementation of ASCVD risk estimation to guide SBP targets described in the 2017 ACC/AHA HTN guidelines with a goal SBP<130/80mm Hg for those at a 10-year ASCVD risk of ≥7.5%, an elevated HF risk identified by PCP-HF tool may be helpful to inform clinician-patient risk discussions regarding intensive blood pressure lowering for HF prevention and identify those patients who may experience a greater relative reduction in HF risk with a lower SBP goal [46]. The PCP-HF tool may also provide the potential for more targeted implementation efforts in those with poorly controlled SBP given recent reports of decreased prevalence of controlled SBP in the US between 2013–2014 and 2017–2018 [47].

The PCP- HF tool also provides a framework that may be useful to test the efficacy of specific anti-hypertensive therapies among individuals at-risk for HF. For example, sacubitril/valsartan therapy, may be a novel anti-hypertensive therapy to be considered in individuals at high risk of HF given evidence of improvement in cardiac mechanics in preclinical and clinical studies [48,49]. A randomized-controlled trial is currently underway to investigate sacubitril/valsartan in the primary prevention of HF among individuals with an elevated natriuretic peptide levels (with a similar strategy to STOP-HF) in the Personalized Prospective Comparison of ARni with ArB in Patients with Natriuretic Peptide eLEvation [50].

In addition to targeted risk factor modification, individuals identified with an intermediate HF risk estimated with the PCP-HF tool may benefit from further risk stratification with sequential screening strategies with biomarkers (e.g. brain natriuretic peptide [BNP]) and/or imaging (e.g. left ventricular hypertrophy on echocardiogram) in an individualized, stepwise approach. The St. Vincent's Screening to Prevent Heart Failure Study (STOP-HF) demonstrated the utility of a biomarker-driven strategy with BNP screening and collaborative cardiovascular care in patients at risk for HF. Use of the PCP-HF tool could further enhance the yield of such a strategy through enriched selection of patients for sequential testing using biomarkers and/or imaging [51]. Future interventions could include surveillance imaging, more aggressive risk factor management, and potential for targeted therapies [52].

## Limitations

There are several limitations to this study. For purposes of parsimonious data presentation, we used interval values for age and risk factors, allowing us to present HF risk across a broad range of values. Many risk factors are correlated; therefore, artificially adjusting one risk factor in isolation is a theoretical application of a multi-variable risk tool and informs understanding of the wide spectrum of potential risk estimates in the population. We were not able to estimate risk by HF subtype (preserved vs. reduced ejection fraction). However, given overlapping risk factors, a combined HF endpoint is meaningful for risk-based communication and targeted prevention strategies. Quantification of HF risk with the PCP-HF tool is limited to short-term risk (10-years) and may provide incomplete risk stratification as individuals at low 10-year risk of HF may be at high lifetime risk of HF, as is observed for the risk of ASCVD in the majority of the population [53]. Further, the PCP-HF tool also does not account for key lifestyle factors that are associated with HF risk, such as dietary patterns including alcohol consumption, sedentary time, cardiopulmonary fitness, or duration of regular physical activity.

Finally, the applicability of the PCP-HF tool in patients who do not identify as White or Black is not well-studied and warrants further investigation.

## Conclusions

The present study provides context to guide clinician-patient risk-based decision making and inform future practice guidelines by demonstrating a wide range of predicted risk values associated with diverse risk factor combinations in White and Black men and women. Use of the heart failure-specific risk score, PCP-HF, may help identify individuals at increased risk for development of HF on a population level and facilitate risk communication on an individual level to reduce the unacceptably high burden of HF.

## Supporting information

**S1 Fig. Calculation of pooled cohort equations to Prevent Heart Failure (PCP-HF).** Data routinely available in ambulatory setting can be entered at www.pcphf.org to calculate race-sex specific ten-year risk of incident HF. BMI = body mass index; HDL = high-density lipoprotein cholesterol.
(TIF)

**S2 Fig. Ten-year predicted heart failure risk by varying cholesterol levels in a hypothetical white and black man.** Ten-year predicted risks for heart failure by varying cholesterol levels in a hypothetical White man (A to B) and Black man (C to D) at selected ages, with other risk factors held constant at approximate age-adjusted national means (including nonsmoking).
(TIF)

**S3 Fig. Ten-year predicted heart failure risk by varying cholesterol levels in a hypothetical white and black woman.** Ten-year predicted risks for heart failure by varying cholesterol levels in a hypothetical White woman (A to B) and Black woman (C to D) at selected ages, with other risk factors held constant at approximate age-adjusted national means (including nonsmoking).
(TIF)

**S4 Fig. Ten-year predicted heart failure risk by varying multiple risk factors at 60 years of age in a hypothetical white and black man.** Ten-year predicted risks for heart failure by varying levels of multiple risk factors in a hypothetical White man (A) and Black man (B) at 60 years of age. BMI, body mass index; BP, blood pressure; DM, diabetes mellitus; SBP, systolic blood pressure.
(TIF)

**S5 Fig. Ten-year predicted heart failure risk by varying multiple risk factors at 60 years of age in a hypothetical white and black woman.** Ten-year predicted risks for heart failure by varying levels of multiple risk factors in a hypothetical White woman (A) and Black woman (B) at 60 years of age. BMI = body mass index, BP = blood pressure, DM = diabetes mellitus, SBP, systolic blood pressure.
(TIF)

**S6 Fig. Ten-Year predicted heart failure risk in various hypothetical patients.** Ten- year predicted risk for heart failure in (A) 60 year old non-smoker with untreated SBP 120 mmHg, treated FG 140 mg/dL, BMI 35 kg/m$^2$, TC = 200mg/dL, HDL-C = 50mg/dL, and QRS 90ms, (B) 60 year old non-smoker with treated SBP 140 mmHg, untreated FG 120 mg/dL, BMI 35 kg/m$^2$, TC = 160 mg/dL, HDL-C = 50mg/dL, and QRS 90ms, (C) 60 year old smoker with untreated SBP 120 mmHg, treated FG 100 mg/dL, BMI 25 kg/m$^2$, TC = 200mg/dL, HDL-C = 50mg/dL, and QRS 90ms and (D) 60 year old non-smoker with treated SBP 140 mmHg

untreated FG 100 mg/dL, BMI 30 kg/m$^2$, TC = 240mg/dL, HDL-C = 40mg/dL, and QRS 90ms. BMI = body mass index; DM = diabetes mellitus; HDL = high-density; FG = fasting glucose; LDL-C = lipoprotein cholesterol; SBP = systolic blood pressure; TC = total cholesterol.
(TIF)

**S7 Fig. Effect of QRS duration on HF Risk in 60 year old with moderate risk factors.** Comparison of ten-year predicted risk for heart failure in a 60 year old of each race-sex group with moderate risk factors with QRS of 90ms or 120ms. Moderate risk factors defined as untreated SBP = 130mmHg; treated FG = 130 mg/dL; BMI = 35kg/ m$^2$; TC = 160 mg/ dL; HDL-C = 40 mg/dL.
(TIF)

**S8 Fig. Comparison of 10-year predicted heart failure risk by race-sex group with QRS of 100ms and 110ms.** Ten-year predicted risk for heart failure for a hypothetical White and Black man and woman at interval selected ages, with risk factors held constant at approximate age-adjusted national means (among nonsmokers). (A) Ten-year heart failure risk estimates for those not taking antihypertensive or diabetes medications with QRS of 100ms. (B) Ten-year heart failure risk estimates for those not taking antihypertensive or diabetes medications with QRS of 110ms.
(TIF)

**S1 Table. Race and sex-specific coefficients from the Ten-year pooled cohort equations to prevent heart failure to calculate predicted risk of heart failure.**
(DOCX)

## Author Contributions

**Conceptualization:** Donald M. Lloyd-Jones, Thanh Huyen T. Vu, Clyde W. Yancy, Sanjiv J. Shah, Mercedes Carnethon, Sadiya S. Khan.

**Data curation:** Aakash Bavishi, Hongyan Ning, Thanh Huyen T. Vu, Mercedes Carnethon, Sadiya S. Khan.

**Formal analysis:** Aakash Bavishi, Hongyan Ning, Thanh Huyen T. Vu, Sadiya S. Khan.

**Funding acquisition:** Sadiya S. Khan.

**Investigation:** Aakash Bavishi, Clyde W. Yancy, Sanjiv J. Shah, Mercedes Carnethon, Sadiya S. Khan.

**Methodology:** Aakash Bavishi, Donald M. Lloyd-Jones, Hongyan Ning, Mercedes Carnethon, Sadiya S. Khan.

**Resources:** Sadiya S. Khan.

**Writing – original draft:** Aakash Bavishi, Donald M. Lloyd-Jones, Sadiya S. Khan.

**Writing – review & editing:** Aakash Bavishi, Donald M. Lloyd-Jones, Hongyan Ning, Thanh Huyen T. Vu, Clyde W. Yancy, Sanjiv J. Shah, Mercedes Carnethon, Sadiya S. Khan.

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
