## [Decision Letter · Decision Letter 0]

17 Aug 2020

PONE-D-20-17009

Systematic examination of a heart failure risk prediction tool: the Pooled Cohort Equations to Prevent Heart Failure

PLOS ONE

Dear Dr. Khan,

Thank you for submitting your manuscript to PLOS ONE. After careful consideration, we feel that it has merit but does not fully meet PLOS ONE’s publication criteria as it currently stands. Therefore, we invite you to submit a revised version of the manuscript that addresses the points raised during the review process.

We look forward to receiving your revised manuscript.

Kind regards,

Antonio Cannatà

Academic Editor

PLOS ONE

Journal Requirements:

2. Please upload a new copy of Figure S3, S4, 6 and 7 as the detail is not clear. Please follow the link for more information: https://blogs.plos.org/plos/2019/06/looking-good-tips-for-creating-your-plos-figures-graphics/"" https://blogs.plos.org/plos/2019/06/looking-good-tips-for-creating-your-plos-figures-graphics/

Additional Editor Comments (if provided):

The manuscript is well written, informative and scientifically solid.

The authors might discuss more in details the potential implication on outcomes.

The authors might consider improving the graphics. Specifically

- a summary figure with a risk chart might be provided

- a figure with the 10-years incidence of HF in representative subjects with a combination of more than 1 risk factor

Please provide a sensitivity analysis using a different cut-offs for the QRS Duration (i.e. 100 and 110 msec)

Reviewers' comments:

Reviewer's Responses to Questions

**Comments to the Author**

1. Is the manuscript technically sound, and do the data support the conclusions?

Reviewer #1: Yes

Reviewer #2: Yes

Reviewer #3: Yes

2. Has the statistical analysis been performed appropriately and rigorously? 

Reviewer #1: Yes

Reviewer #2: Yes

Reviewer #3: Yes

3. Have the authors made all data underlying the findings in their manuscript fully available?

Reviewer #1: Yes

Reviewer #2: Yes

Reviewer #3: Yes

4. Is the manuscript presented in an intelligible fashion and written in standard English?

Reviewer #1: Yes

Reviewer #2: Yes

Reviewer #3: Yes

5. Review Comments to the Author

Reviewer #1: The overal impression is good. I see it as a demonstration of the effects of the components of the validated PCP-HF risk prediction tool. I can accept the statement of the significant influence of age, race and gender, since this has been reported in other epidemiological heart failure studies. However, I cannot clearly abstract from the article, if all used variables imply causality for incident heart failure, and if not, what underlying phenomena might they be representing. I would appreciate this in the discussion, since preventive strategies should be based on causal factors. Also I missed pointing out the phenomenon of reverse epidemiology considering BMI in white males of advanced age. Why do higher BMI levels have lower incidences, but only in white males? Another limitation of the implementation in a preventive programme I percieve ist he case of mixed racial ancestry and lack of information about alcohol consumption and physical activity. To sum up, I think this is a valid and inspirative paper witch points many important considerations in search of causality and individually tailored HF-prevention. The authors may consider adressing a gently brighter view of the limitations and discussing some outstanding factor combinations.

Reviewer #2: The objective of the study of Aakash Bavishi and collegues was to systematically evaluate the PCP-HF risk prediction tool to determine the impact of varying risk factor levels and combinations in different age, sex, and race groups. As underlined by Authors, the study provides important informations to guide clinician-patient risk-based decision making and for future practice guidelines by demonstrating a wide range of predicted risk values associated with diverse risk factor combinations in white and black men and women. The use of PCP-HF score may help identify individuals at increased risk for development of HF on a population level and facilitate risk communication on an individual level to reduce the unacceptably high burden of HF.

Tha paper is well presented, complete, clear. In my opinion could be accepted as is.

Reviewer #3: The study addresses the utility of a risk prediction model for heart failure (Combining Cohort Equations to Prevent IC-PCP-HF) to demonstrate the range of risk values in combination with race-sex group.

The authors previously published the PCP-HF tool, which incorporates age, systolic blood pressure, fasting blood glucose, body mass index, total cholesterol, HDL-c, smoking, use of antihypertensive drugs, use of diabetes mellitus drugs and QRS duration in multivariate Cox proportional risk regression equation to predict the absolute 10-year risk for incident HF.

This manuscript does an excellent job demonstrating how differences in sex and ethnicity, in addition to the combination of various risk factors, can predict the risk of heart failure in 10 years.

As proposed by the authors, the ability to predict heart failure is desirable to help physicians to treat risk factors.

6. PLOS authors have the option to publish the peer review history of their article (what does this mean?). If published, this will include your full peer review and any attached files.

Reviewer #1: No

Reviewer #2: **Yes: **Andrea Di Lenarda, MD, FESC, FACC

Reviewer #3: No

---

## [Author Response · Author response to Decision Letter 0]

22 Sep 2020

1) The authors might discuss more in details the potential implication on outcomes 

Authors’ Response: We thank the editor for this recommendation and agree that this discussion warrants more details hypothesizing potential clinical benefits of risk-based prevention of heart failure with a heart failure-specific tool. In addition, we appreciate the opportunity to highlight emerging clinical interventions, such as sodium glucose co-transporter 2 inhibitors and sacubitril/valsartan, that have a widening evidence base to prevent heart failure. We have added and edited the following statements in our discussion to provide more detail: 

• On Page 11, lines 161-163, “Implementation of HF-specific risk tools may facilitate preventive efforts, reduce incidence of HF, and improve cardiovascular outcomes in the population”

• On Page 12, lines 178-183, “Risk-based randomized control trials to further investigate the efficacy of SGLT2i in the primary prevention of HF among patients with and without diabetes are needed and may inform novel patient populations who may benefit. Use of the PCP-HF tool can also be applied to enrich clinical trial enrollment by focusing on at-risk individuals to investigate novel therapies in a targeted manner to broaden risk-based prevention interventions.” 

• On Page 12, lines 193 - 195, “The PCP-HF tool may also provide the potential for more targeted implementation efforts in those with poorly controlled SBP given recent reports of decreased prevalence of controlled SBP in the US between 2013-2014 and 2017-2018.”

1. Muntner P, Hardy ST, Fine LJ, Jaeger BC, Wozniak G, Levitan EB, Colantonio LD. Trends in blood pressure control among US adults with hypertension, 1999-2000 to 2017-2018. JAMA. 2020.

2) The authors might consider improving the graphics. Specifically

- a summary figure with a risk chart might be provided

- a figure with the 10 years-incidence of HF in representative subjects with a combination of more than 1 risk factor

Authors’ Response: We appreciate the Editor’s suggestions to improve the graphic presentation of our results. We have revised Figure 1 so that it now includes a risk chart demonstrating ten-year predicted risk for HF for a hypothetical white and black man and woman at selected ages. We have also added a Supplemental Figure 1 which summarizes the components of the PCP-HF risk tool and online interface. We have also created an additional Supplemental Figure 6 which shows 10 year incident HF risk in each race-sex group for different risk factor combinations commonly seen in clinical practice for a hypothetical 60 year old. 

3) Please provide a sensitivity analysis using different cut-offs for the QRS duration(100 and 110 msec) 

Authors’ Response: We thank the Editor for this suggestion. We have added a Supplemental Figure 7 which depicts the effects of prolonging QRS from 90ms to 120ms in a 60 year old with moderate risk factor levels. We have also created a Supplemental Figure 8 which compares 10-year predicted HF risk by race-sex group at selected ages in normotensive patients without diabetes and varying QRS of 100ms vs 110ms. 

Reviewer 1 Comments to the Authors

1. The overall impression is good. I see it as a demonstration of the effects of the components of the validated PCP-HF risk prediction tool. I can accept the statement of the significant influence of age, race and gender, since this has been reported in other epidemiological heart failure studies. However, I cannot clearly abstract from the article, if all used variables imply causality for incident heart failure, and if not, what underlying phenomena might they be representing. I would appreciate this in the discussion, since preventive strategies should be based on causal factors. 

Authors’ Response: We thank the reviewer for requesting this important clarification. We agree that distinguishing causality from associations is important in the translation towards targeted preventive strategies. All of the components of the PCP-HF risk tool: systolic blood pressure, fasting glucose, presence of hypertension and/or diabetes mellitus, total cholesterol, high-density lipoprotein cholesterol, body mass index, smoking, and QRS duration are associated with incident HF. Of these, prolonged QRS duration, is likely not causal and reflects evidence of subclinical cardiac remodeling as a surrogate marker of heart failure risk. Available data supporting benefits in reduction of heart failure risk have focused on weight management/loss, diabetes and hypertension control, statin therapy, and smoking cessation. We agree that preventive strategies should focus on these causal factors and have updated the discussion:

• On Page 9, lines 127-131 “The previously validated components of the PCP-HF tool were selected based on their individual associations with incident HF and reflect causal factors that contribute to HF risk (obesity, diabetes, hypertension, hyperlipidemia, and smoking) as well as subclinical surrogates of HF risk (QRS duration). For example, higher levels of SBP were associated with greater predicted risk of HF in all race-sex groups.”

1.Dhingra R, Pencina MJ, Wang TJ, Nam B-H, Benjamin EJ, Levy D, et al. Electrocardiographic QRS duration and the risk of congestive heart failure: the Framingham Heart Study. Hypertension (Dallas, Tex : 1979). 2006;47(5):861-7.

2. Velagaleti RS, Massaro J, Vasan RS, Robins SJ, Kannel WB, Levy D. Relations of lipid concentrations to heart failure incidence: the Framingham Heart Study. Circulation. 2009;120(23):2345-51.

2. Also I missed pointing out the phenomenon of reverse epidemiology considering BMI in white males of advanced age. Why do higher BMI levels have lower incidences, but only in white males? 

Authors’ Response: We thank the reviewer for pointing out this interesting detail.. As you know, there is significant controversy regarding the obesity paradox broadly in cardiovascular disease, but also in heart failure, specifically. To highlight and clarify this point, we have added the following to our discussion section of the manuscript: 

• On Page 10, lines 133-142, “While higher BMI values were generally associated with greater predicted risk of HF, there was an inverse association between BMI and predicted risk of HF in the theoretic risk curve for a 70-year old White man. One potential explanation may be the controversial obesity paradox in which patients with obesity with risk factors for HF such as HTN or DM paradoxically had better outcomes in terms of mortality when compared with patients in lower BMI strata. Specifically, among older adults with obesity, there are several explanations for this phenomenon, including a greater competing risk of mortality from non-HF causes, muscle wasting and. sarcopenia, as well as malnutrition or development of cachexia. Finally, BMI is a poorer measure of adiposity in white men and may reflect greater burden of frailty compared with other race-sex groups.”

1. Zoppini G, Verlato G, Leuzinger C, Zamboni C, Brun E, Bonora E, et al. Body mass index and the risk of mortality in type II diabetic patients from Verona. International journal of obesity and related metabolic disorders : journal of the International Association for the Study of Obesity. 2003;27(2):281-5.

2. Jackson CL, Yeh H-C, Szklo M, Hu FB, Wang N-Y, Dray-Spira R, et al. Body-Mass Index and All-Cause Mortality in US Adults With and Without Diabetes. J Gen Intern Med. 2014;29(1):25-33.

3. Shah RV, Abbasi SA, Yamal JM, Davis BR, Barzilay J, Einhorn PT, et al. Impaired fasting glucose and body mass index as determinants of mortality in ALLHAT: is the obesity paradox real? The Journal of Clinical Hypertension. 2014;16(6):451-8.

4. Barrett-Connor E, Khaw K. Is hypertension more benign when associated with obesity? Circulation. 1985;72(1):53-60.

5. Oreopoulos A, Kalantar-Zadeh K, Sharma AM, Fonarow GC. The obesity paradox in the elderly: potential mechanisms and clinical implications. Clinics in geriatric medicine. 2009;25(4):643-59, viii.

6. Burkhauser RV, Cawley J. Beyond BMI: the value of more accurate measures of fatness and obesity in social science research. Journal of health economics. 2008 Mar 1;27(2):519-29.

3. Another limitation of the implementation in a preventive programmed I perceive is the case of mixed racial ancestry and lack of information about alcohol consumption and physical activity. 

Authors’ Response: We thank the reviewer for raising this important point. We agree that this is a limitation of our manuscript. We have therefore adjusted our manuscript as below. 

• On Page 14, lines 225-229, “The PCP-HF tool also does not account for key lifestyle factors that are associated with HF risk, such as dietary patterns including alcohol consumption, sedentary time, cardiopulmonary fitness, or duration of regular physical activity. Finally, the applicability of the PCP-HF tool in patients who do not identify as White or Black is not well-studied and warrants further investigation.”

4. To sum up, I think this is a valid and inspirative paper which points many important considerations in search of causality and individually tailored HF-prevention. The authors may consider addressing a gently brighter view of the limitations and discussing some outstanding factor combinations.

Authors’ Response: We thank the reviewer for their positive comments on our manuscript. 

Reviewer 2 Comments to the Authors

The objective of the study of Aakash Bavishi and collegues was to systematically evaluate the PCP-HF risk prediction tool to determine the impact of varying risk factor levels and combinations in different age, sex, and race groups. As underlined by Authors, the study provides important information to guide clinician-patient risk-based decision making and for future practice guidelines by demonstrating a wide range of predicted risk values associated with diverse risk factor combinations in white and black men and women. The use of PCP-HF score may help identify individuals at increased risk for development of HF on a population level and facilitate risk communication on an individual level to reduce the unacceptably high burden of HF. The paper is well presented, complete, clear. In my opinion could be accepted as is.

Authors’ Response: We thank the reviewer for their gracious review of our manuscript. 

Reviewer 3 Comments to the Authors

The study addresses the utility of a risk prediction model for heart failure (Combining Cohort Equations to Prevent IC-PCP-HF) to demonstrate the range of risk values in combination with race-sex group. The authors previously published the PCP-HF tool, which incorporates age, systolic blood pressure, fasting blood glucose, body mass index, total cholesterol, HDL-c, smoking, use of antihypertensive drugs, use of diabetes mellitus drugs and QRS duration in multivariate Cox proportional risk regression equation to predict the absolute 10-year risk for incident HF.

This manuscript does an excellent job demonstrating how differences in sex and ethnicity, in addition to the combination of various risk factors, can predict the risk of heart failure in 10 years. As proposed by the authors, the ability to predict heart failure is desirable to help physicians to treat risk factors.

Authors’ Response: We thank the reviewer for their thoughtful summary and positive review of our manuscript.

---

## [Editor Report · Decision Letter 1]

29 Sep 2020

Systematic examination of a heart failure risk prediction tool: the Pooled Cohort Equations to Prevent Heart Failure

PONE-D-20-17009R1

Dear Dr. Khan,

We’re pleased to inform you that your manuscript has been judged scientifically suitable for publication and will be formally accepted for publication once it meets all outstanding technical requirements.

Kind regards,

Antonio Cannatà

Academic Editor

PLOS ONE

Additional Editor Comments (optional):

The manuscript is now scientifically improved and meets the publication criteria of PLOS ONE.
---

## [Editor Report · Acceptance letter]

5 Oct 2020

PONE-D-20-17009R1 

Systematic examination of a heart failure risk prediction tool: the Pooled Cohort Equations to Prevent Heart Failure 

Dear Dr. Khan:

I'm pleased to inform you that your manuscript has been deemed suitable for publication in PLOS ONE. Congratulations! Your manuscript is now with our production department. 

Kind regards, 

on behalf of

Dr. Antonio Cannatà 

Academic Editor

PLOS ONE